# Is Homomorphic Encryption-Based Deep Learning Secure Enough?

**DOI:** 10.3390/s21237806

**Published:** 2021-11-24

**Authors:** Jinmyeong Shin, Seok-Hwan Choi, Yoon-Ho Choi

**Affiliations:** School of Computer Science and Engineering, Pusan National University, Busan 609-735, Korea; sinryang@pusan.ac.kr (J.S.); daniailsh@pusan.ac.kr (S.-H.C.)

**Keywords:** deep learning, privacy-preserving, homomorphic encryption, adversarial examples, reconstruction attack, membership inference attack

## Abstract

As the amount of data collected and analyzed by machine learning technology increases, data that can identify individuals is also being collected in large quantities. In particular, as deep learning technology—which requires a large amount of analysis data—is activated in various service fields, the possibility of exposing sensitive information of users increases, and the user privacy problem is growing more than ever. As a solution to this user’s data privacy problem, homomorphic encryption technology, which is an encryption technology that supports arithmetic operations using encrypted data, has been applied to various field including finance and health care in recent years. If so, is it possible to use the deep learning service while preserving the data privacy of users by using the data to which homomorphic encryption is applied? In this paper, we propose three attack methods to infringe user’s data privacy by exploiting possible security vulnerabilities in the process of using homomorphic encryption-based deep learning services for the first time. To specify and verify the feasibility of exploiting possible security vulnerabilities, we propose three attacks: (1) an adversarial attack exploiting communication link between client and trusted party; (2) a reconstruction attack using the paired input and output data; and (3) a membership inference attack by malicious insider. In addition, we describe real-world exploit scenarios for financial and medical services. From the experimental evaluation results, we show that the adversarial example and reconstruction attacks are a practical threat to homomorphic encryption-based deep learning models. The adversarial attack decreased average classification accuracy from 0.927 to 0.043, and the reconstruction attack showed average reclassification accuracy of 0.888, respectively.

## 1. Introduction

According to deep learning, which shows impressive performance in many fields, deep learning models are applied to many services that use large amounts of data related to safety and sensitive information, such as autonomous driving, health care and finance. However, many researches showed that deep learning has vulnerabilities in performance [1,2,3] and privacy [4,5,6]. For example, increasing false positives and false negatives of practical deep learning system such as face recognition and object recognition system using adversarial examples [2,7,8,9] and extracting face or sensitive medical images using deep learning privacy attacks [5,10].

Homomorphic encryption which enables arithmetic (or logical) operations on encrypted data is one of the strong candidates to prevent such attacks. By preventing unauthorized access from an adversary who does not have a proper key, homomorphic encryption can separate the adversary from deep learning models. Moreover, since modern homomorphic encryption schemes are constructed based on the lattice cryptosystem, the homomorphic encryption provides quantum resistance. Therefore, many researchers have endeavored to apply homomorphic encryption to deep learning as a sustainable countermeasure of attacks for deep learning models and there are various milestone works, such as *Cryptonets* [11], *CryptoDL* [12] and *He-transformer* [13].

However, when homomorphic encryption-based deep leaning model is deployed as a service, we cannot ensure that homomorphic encryption can protect deep learning model from adversaries. As state of the art practical homomorphic encryption schemes use one secret key to decrypt encrypted data, a trusted party who keeps the secret key and delivers decrypted results should be included in the service model. Thus, the communication between client and trusted party cannot be covered by homomorphic encryption perfectly. Moreover, since the homomorphic encryption-based deep learning model communicates with clients using paired input and output data, an adversary can perform black-box based attack which does not require any information about the model. In addition to this, when considering multi-party computation scenario that some secret information is shared to others, attacks by insider are much easier.

To verify such vulnerabilities, we propose three attack models. The first one handles the threat of adversarial examples in communication between client and trusted party. Second one handles the threat of reconstruction attack in case that homomorphic encryption-based deep learning service is open to everyone. The last one handles threat of membership inference attack by semi-honest party who behaves honestly but is curious about other party’s information in multi-party computation situation. In addition, we show that such attack models are feasible by performing attacks on homomorphic encryption-based deep learning models.

Main contributions of this paper can be summarized as follows:We propose three specific and practical attack models for homomorphic encryption-based deep learning model using adversarial example, reconstruction attack and membership inference attack. Especially, one of them is related to model performance which have not been considered as a threat to homomorphic encryption-based applications.We show the feasibility of our attack model by performing attacks on homomorphic encryption-based deep learning models directly. To the best of our knowledge, this work is the first work that verifies the attack model on homomorphic encryption-based application with real attacks.

The rest of the paper is organized as follows. In Section 2, we explain background knowledge about attacks on deep learning and homomorphic encryption. Then, we propose our three attack models for homomorphic encryption-based deep learning in Section 4. In Section 5, we show the feasibility of our attack model by performing attacks on homomorphic encryption-based deep learning model. Finally, we summarize this paper in Section 6.

## 2. Background

### 2.1. Attacks on Deep Learning Based Services

As deep learning is widely applied to sensitive services such as autonomous driving, health care, and finance, adversaries have tried to exploit such services using characteristic of deep learning. In this section, we briefly introduce two type of threats on deep learning model, (1) adversarial example; and (2) privacy attacks;.

**Adversarial Example:** An adversarial example is a human-imperceptible perturbations added image, which is misclassified by machine. The adversarial attack is a process to generate human-imperceptible perturbations and there are three well-known adversarial attack methods.

The Fast Gradient Sign Method (FGSM) is a non-iterative-based fast adversarial perturbation generation method proposed by Goodfellow et al. [1]. This method generates perturbation to increase loss of deep learning models using sign of gradients.

The Basic Iterative Method (BIM) is a basic iterative adversarial perturbation generation method proposed by Kurakin et al. [2]. In contrast to FGSM which updates gradients once, BIM updates gradients many times to optimize perturbation in fine manner.

C&W is an $L_{0}$, $L_{\infty }$ and $L_{2}$ distance based iterative attack proposed by Carlini and Wagner [3]. C&W shows a higher attack success rate than other methods while minimizing the magnitude of perturbation. In this paper, we consider the $L_{2}$ type of C&W attack only, which is most frequently mentioned in other works [14,15]

**Privacy Attacks on Neural Network:** A privacy attacks on a neural network is a threat that extracts any sensitive information without complete data. Mainly, sensitive data is extracted from deep learning model using specific privacy attacks. According to type of target information, these attacks can be categorized into: (1) reconstruction attack; (2) membership inference attack; (3) property inference attack; and (4) model extraction attack. In this paper, we mainly address reconstruction and membership inference attacks whereby the critical attack model can be defined in encryption based privacy-preserving deep learning models.

A reconstruction attack aims to restore target information given partial information such as insensitive information or label. Attacks of this category were first introduced by Fredrikson et al. and can be grouped into two sub-categories, attribute inference(partial data reconstruction) [4] and model inversion (full data reconstruction) [5]. Attribute inference attack infers sensitive information using probabilistic technique such as Maximum A Posteriori (MAP) when distribution of sensitive information and insensitive information of target data are known. Model inversion attack reconstructs training samples respective their labels when iterative queries on target model are possible. Since intuition of model inversion attack is based on gradient descent technique, the majority of model inversion attacks are implemented on white-box setting to use gradients information of target model [16,17,18]. To handle the limitation of a white-box setting, non gradient based attacks using generative network or Generative Adversarial Network (GAN) are also proposed [19,20].

Membership inference attack aims to determine whether an input data x was used during the training phase of the target model. Shokri et al. [6] proposed the first membership inference attack which uses output vectors of target model only (black-box setting). After [6], Nasr et al. [21] showed that membership inference attack can be performed more effectively in collaborative setting where model parameters and gradients are known(white-box setting).

### 2.2. Homomorphic Encryption

Homomorphic Encryption (HE) is an encryption scheme which can perform some additional evaluations over encrypted data. Given a secret key sk, a public key pk, an encryption function E(), a decryption function D() and two plain text ${\mathrm{pt}}_{1}$ and ${\mathrm{pt}}_{2}$, following equation can be defined in a HE.
(1)$$D\left(E\left({\mathrm{pt}}_{1},pk\right)⊗E\left({\mathrm{pt}}_{2},pk\right),sk\right)=D\left(E\left({\mathrm{pt}}_{1}\times {\mathrm{pt}}_{2},pk\right),sk\right)$$

In Equation (Equation 1), an encryption scheme S(E,D,⊗,sk,pk) is homomorphic encryption scheme if and only if there is an operation ⊗ corresponding to the operation × which is defined in plain text domain.

The latest HE scheme called Fully Homomorphic Encryption (FHE) was first introduced by C. Gentry [22]. Gentry’s FHE scheme removes the limitation of the number of operations by introducing bootstrapping operation. Based on Gentry’s FHE scheme, recent FHE schemes, such as Brakerski–Gentry–Vaikuntanathan (BGV) [23], Brakerski–Fan–Vercauteren–Vaikuntanathan (BFV) [24] and Cheon–Kim–Kim–Song(CKKS) [25], have improved their performance to a practical level by optimizing the lattice structure and operations for the Ring Learning with Error (RLWE) problem. In addition, since recent FHE schemes, including Gentry’s FHE scheme, are designed based on the lattice encryption scheme considering the quantum computing environment, modern FHE schemes are considered as strong candidates for sustainable privacy-preserving computation technique after the quantum computing era.

## 3. Related Works

### 3.1. Privacy-Preserving Deep Learning Using HE

Since HE schemes can prevent unauthorized access from anyone who does not a have proper key, the adversary is perfectly separated from sensitive data and the operation process of deep learning model. Thus, after modern FHE schemes have improved their performance to practical level, many researchers are endevoured to apply homomorphic encryption to deep leaning to achieve privacy-preserving deep learning.

Gilad-Bachrach et al. [11] proposed *Cryptonets*, the first neural network based on homomorphic encryption including reasonable verification. They designed a very small neural network to avoid bootstrapping operation and used square operation instead of other activation functions.

Hesamifard et al. [12] proposed *CryptoDL* that focused on linear approximation of ReLU activation function. They increased test accuracy on cifar-10 dataset to 91.5% using batch normalization and approximated activation function calculated from the integrated sigmoid function. However, since their model architecture does not consider overhead from HE operations and bootstrapping, the inference speed of the model is impractical level.

Boemer et al. [26,27,28] proposed *HE-transformer* that the first graph compiler for artificial neural networks which has HE backend. They implemented HE-transformer by integrating Intel’s *nGraph* [29] compiler with Microsoft’s *SEAL* [30] library. Although HE-transformer does not provide optimal implementation of HE-based neural network, it is meaningful work in that high throughput HE-based neural network can be easily implemented and tested.

### 3.2. Attacks on Privacy-Preserving Deep Learning

As many privacy-preserving deep learning methods are introduced, some researchers have proposed attack methods targeting privacy-preserving deep learning models to verify the security of the existing privacy-preserving deep learning or to show inadequacy its as a service.

Rahman et al. [31] tried to exploit differential privacy based privacy-preserving deep learning model using membership inference attack proposed by Shokri et al. [6]. They showed that differential privacy based stochastic gradient descent method decreases the attack success rate meaningfully. However, since their attack scenario requires model information such as model structure and learning parameters, this scenario is not applicable to HE-based deep learning models where the adversary cannot acquire model information.

Chang et al. [32] proposed reconstruction attack methods targeting machine perceptible image encryption based privacy-preserving deep learning model. Although machine perceptible image encryption results are human imperceptible, the encrypted image contains context of original image to be classified by machines. Thus, Chang et al. proposed reconstruction algorithm that re-arranges the pixels of encrypted image using the context of the original image. However, since recent HE schemes provide *semantic security*, it is almost impossible to get a general context of the original data from the encrypted data. Therefore, their methods cannot be applied to HE-based deep learning models.

Wu et al. [33] proposed an attack method against HE-based deep learning. In this method, the adversary must be the participant of federated learning using HE. When an adversary trains a model for the first time, the adversary saves the current model parameters. After processing a few steps, the adversary exploits the model by uploading the difference between current model parameters and saved model parameters to reset the training results of other parties. Such an attack can easily reduce the accuracy of the model, but the reduced accuracy can be easily found before deployed as service. Therefore, their method is not practical threat to services using HE-based deep learning models.

Different from the existing attacks that cannot be applied to HE-based deep learning model or that can be detected before service deployment, we propose three attacks that consider the vulnerabilities of the practical HE-based deep learning service scenario. The first two attacks exploit services using HE-based deep learning which is already deployed. In such scenarios, an adversary can exploit target model without access to the key to decrypt data which is encrypted using the HE scheme. The last attack, membership inference attack, does not interact with other parties during attack phase, making it difficult to detect the attack before the target system is exploited.

## 4. Attack Model

Since data is encrypted by the client and calculated in the encrypted state, adversary has no chance of accessing any data. As a result, the HE-based deep learning models seems to have immunity on attacks targeting deep learning unless the secret key is not spoiled to adversary. However, when considering the structure of HE schemes and the application scenario of HE-based deep learning models, the model or data can be exploited using existing attack methods. In this section, we introduce three attack model using adversarial example, reconstruction attack and membership inference attack, respectively.

### 4.1. Adversarial Example

Since no HE schemes support multiple key structure that one can encrypt data with own secret key and calculate the data with a model encrypted with another key in practical level, the service model using HE-based deep learning should include trusted third party who preserve secret key. As shown in Figure 1a, A client sends sensitive data to trusted party that has decryption key for result of HE-based deep learning model using secure channel. Then, the trusted party encrypts data using encryption key and calculate requested operation using encrypted data on untrustable environment (e.g., cloud server). After calculation is done, the trusted party decryptes the encrypted result and sends decrypted result to the client. This service model is just same as usual cloud service in perspective of client. Therefore, this kind of scenarios are always exposed to Man-In-The-Middle (MITM) attack. As shown in Figure 1b, the exploited client sends their sensitive data to the adversary. Then, the adversary modifies the data using the adversarial attack method and sends it to trusted party. Since the trusted party cannot recognize the modification of the original data, the trusted party processes the received data as in a normal scenario. As a result, the client receives an abnormal result and makes the wrong decision.

As mentioned in our previous work, adversarial attacks on cloud-like deep learning service models can cause safety issues [34]. In addition to this, we can consider a scenario in finance system. Let us assume that a bank uses the result of HE-based deep learning model when decides whether to approve a loan. If you want to get a loan, the bank has to send your information, such as identification, incomes, etc., to service provider of HE-based deep learning. During this process, adversary can intercept sensitive information and change it slightly not to approve the loan. Since HE-based deep learning service provider get your data from secure channel and the value is not abnormal, it is hard to find these kind of modification and you will not get a loan. Thus, adversarial example is not only a threat to normal deep learning models, but also to HE-based deep learning models.

### 4.2. Reconstruction Attack

Since HE-based deep learning models does not reveal any information except their input and output, the majority of reconstruction attacks which require gradient (weight) information of target model is not a threat. However, black-box reconstruction attacks [19,20] can be a threat without any modification in HE-based deep learning model environment. Thus, compared to other attacks, adversary can exploit a HE-based model using reconstruction attack easily. As shown in Figure 2, the only condition of adversary is access to the model. If the adversary can send a query to the HE-based deep learning model and get a result from the query, the black-box methods can perform reconstruction attack and reveal the sensitive information in training data.

For example, when an adversary performs Fredrikson et al.’s black-box reconstruction attack [5], the adversary sends an arbitrary image to the server and gets result from the server. After comparing difference between target label and result, the image is updated using gradient descent method. By repeating such steps, the adversary can reconstruct an image of the target label.

Since reconstruction attack recovers whole data itself, it can spoil critical privacy from data. For example, in face recognition system, effective reconstruction attack recovers whole face of a human using label data, which is equivalent to name, only [5]. Furthermore, in X-ray recognition system, another disease information that patient does not want others to know is extracted from reconstructed X-ray image.

### 4.3. Membership Inference Attack

Similar to a reconstruction attack, the membership inference attack requires some model information such as model architecture and hyper-parameters. Therefore, a membership inference attack cannot be performed in usual situations. However, when we consider a semi-honest party in the training phase that is operating honestly but curious about others data, a membership inference attack can be performed. As shown in Figure 3, when three parties including one semi-honest party train a model, these parties share all information about the model except their own data and train the model when all parties agree with the model. In this scenario, the semi-honest party can gain access to all the information needed to perform a membership inference attack. After the training is completed, the semi-honest party can obtain extra data including some data similar to the data used in the training phase. As the semi-honest party already knows the model information, they can generate the most desirable shadow model, which reproduces the target model’s behavior and performs an ideal membership inference attack.

For example, when a semi-honest adversary performs Shokri et al.’s membership inference attack [6], the adversary can construct the ideal shadow model because the model parameters and architecture are already shared to all parties. After constructing the shadow model, the adversary trains the shadow model and an attack model using their own data. Then, the adversary can analyze the classification result of arbitrary data to perform a membership inference attack. Furthermore, the membership inference attack performed by the adversary cannot be detected since there is no interaction with other parties.

The membership inference attack which extracts the membership information only is not a critical attack in itself. However, since each party knows additional information about the others, membership information can be linked with this background knowledge and used to identify someone’s identity and sensitive information.

## 5. Experiments

To experiment with the feasibility of our three attack models, we performed experiments using MNIST image classficiation dataset [35] which consists of 60,000 training images and 10,000 testing images corresponding to 10 classes and Fashion-MNIST image classification dataset [36] which consists of 60,000 training images and 10,000 testing images corresponding to 10 classes.

All experiments are implemented on the Ubuntu 20.04 LTS which provided Ubuntu official repository on Docker Hub, 3.80 GHz CPU clock(AMD Ryzen 9 3900XT 12-Core Processor), 128 GB RAM and two Nvidia GeForce RTX 2080 Ti environment.

### 5.1. Implementation

In this section, we explain the target model architecture and implementation method of attack algorithms in detail. Every target model in this section is implemented using IntelAI’s He-transformer [13].

#### 5.1.1. Target Model

Both target models are designed based on He-transformer’s reference Cryptonets implementation. Although He-transformer’s reference Cryptonets shows enough performance for our experiments, too many layers and square activation functions lead to gradient overflow with high probability. So, we modified Cryptonets, as shown in Table 1 and Table 2.

Target model for MNIST dataset consists of 6 layers including one convolution layer, two square activations, one hidden dense layer and output layer. Target model for Fashion-MNIST dataset has much simple structure including one convolution layer, one square activation, one hidden dense layer and output layer. The baseline accuracy of the target models are shown in Table 3.

#### 5.1.2. Adversarial Example

Adversarial attack methods to generate adversarial examples are implemented using *CleverHans* v4.0.0 library [37]. The CleverHans library currently supports FGSM, BIM, C&W attack methods.

#### 5.1.3. Reconstruction Attack

The reconstruction attack is implemented based on He et al.’s black-box reconstruction attack [19]. To extract inter-layer features, He et al.’s attack assumes a collaborative inference model whereby each layer of the model is computed in a different environment. However, since all inter-layer calculation results are encrypted, the adversary cannot extract inter-layer features from the model. Considering such limitation, we modified He et al.’s attack method to using the output of target model only and the modified version of reconstruction attack is shown in Algorithm 1. The function BlackBoxAttack() shows overall operation of black-box reconstruction attack. First, the training dataset for generator is set to a dataset which has similar distribution with training dataset of the target model(line 4). Then, generator *G* is trained with these dataset(line 5). After training of generator is done, generator *G* generates output x^0 which is almost equivalent to original data $x_{0}$. Most of generator training process(line 9-18) is similar to the He et al.’s method. However, to replace inter-layer features with the output of target model, we modified the loss calculation method as shown in line 13.
**Algorithm 1** Reconstruction Attack1:**Function** BlackBoxAttack(*f*, $f(x_{0})$)2:/* *f*: target model*/3:/* $f(x_{0})$: output of training data $x_{0}$ */4:X←**test-dataset**5:G←TrainInverseGenerator(*X*, *f*)6:x^0←$G(f\left(x_{0}\right))$7:**return** x^08: 9:**Function** TrainInverseGenerator(*X*, *f*)10:$G^{\left(0\right)}\leftarrow$**generate_model**()11:**while** t<T**do**12:    **for** *x*
**in**
*X* **do**13:        $L\left(G^{\left(t\right)}\right)\leftarrow {\left|G^{\left(t\right)}\left(f\left(x\right)\right)-x\right|}_{2}^{2}$14:        $G^{\left(t+1\right)}\leftarrow G^{\left(t\right)}-\alpha \frac{\partial L(G^{\left(t\right)})}{G^{\left(t\right)}}$15:    **end for**16:    t←t+117:**end while**18:**return** $G^{\left(T\right)}$

#### 5.1.4. Membership Inference Attack

Since our attack model on membership inference attack assumes the best scenario for Shokri et al.’s membership inference attack method [6], we did not modify their method. So, we just implemented Shokri et al.’s membership inference attack method to suit our environment.

### 5.2. Adversarial Example

To verify feasibility of our first attack model, we performed adversarial attacks, such as FGSM, BIM, C&W, on baseline target model with 0.3 of epsilon value. As shown in Table 4, the accuracy of target model decreased from 0.971 to 0.081, 0.017, 0.027 by FGSM, BIM, C&W, respectively, in MNIST dataset and from 0.884 to 0.094, 0.013, 0.031 by FGSM, BIM, C&W, respectively, in Fashion-MNIST dataset. As a result, all attacks method show dramatic accuracy decreasing of both of target model. Since HE-based deep learning model only blocks access of unauthorized user who does not have proper key, there are no other defence methods against to adversary when the key is spoiled. However, as mentioned in our attack model, adversary can exploit service without any key. Thus, HE-based deep learning model should have defense method, such as adversarial training, before deployed in service.

### 5.3. Reconstruction Attack

To verify the feasibility of our second attack model, we performed reconstruction attack, which is mentioned on Section 5.1.3, to trained target models. Each attack was performed using 10,000 dataset partitioned from each model’s training data, i.e., test data. To show the overall attack success rate, we measured re-classification accuracy, the classification accuracy of reconstructed data which used in Fredrikson et al.’s reconstruction attack [5], and *Structural Similarity* (SSIM) which calculates the similarity between the original image and the reconstructed image [38]. Table 5 shows the measured reconstruction results of each model. The base-line accuracy refers to the classification accuracy of the original images that is the target of reconstruction attack. As shown in Table 5, the re-classification accuracy shows decreased from 0.999 and 0.941 to 0.945 and 0.832 for each dataset, respectively. However, the average SSIM of the reconstructed image showed impressive scores of 0.998 and 0.972, respectively.

Figure 4 shows reconstruction results of each model. As shown in Figure 4a, most of the reconstruction results on MNIST dataset are very similar to each original image, respectively. However, some special cases, such as the third and fourth images in Figure 4a, can be recognized as same number of the original image but cannot reconstruct similarly to the original image. In case of Fashion-MNIST dataset, which is shown in Figure 4b, we can see the overall shape of object is reconstructed well but details of each object are missing.

However, when considering other state-of-the-art black-box reconstruction attacks showing similar result that details are missing on reconstructed image according to the complexity of image [19,20], it seems that the reconstruction attack can be a threat in HE-based deep learning model environment with the development of black-box based attack method.

**Figure 4 sensors-21-07806-f004:**
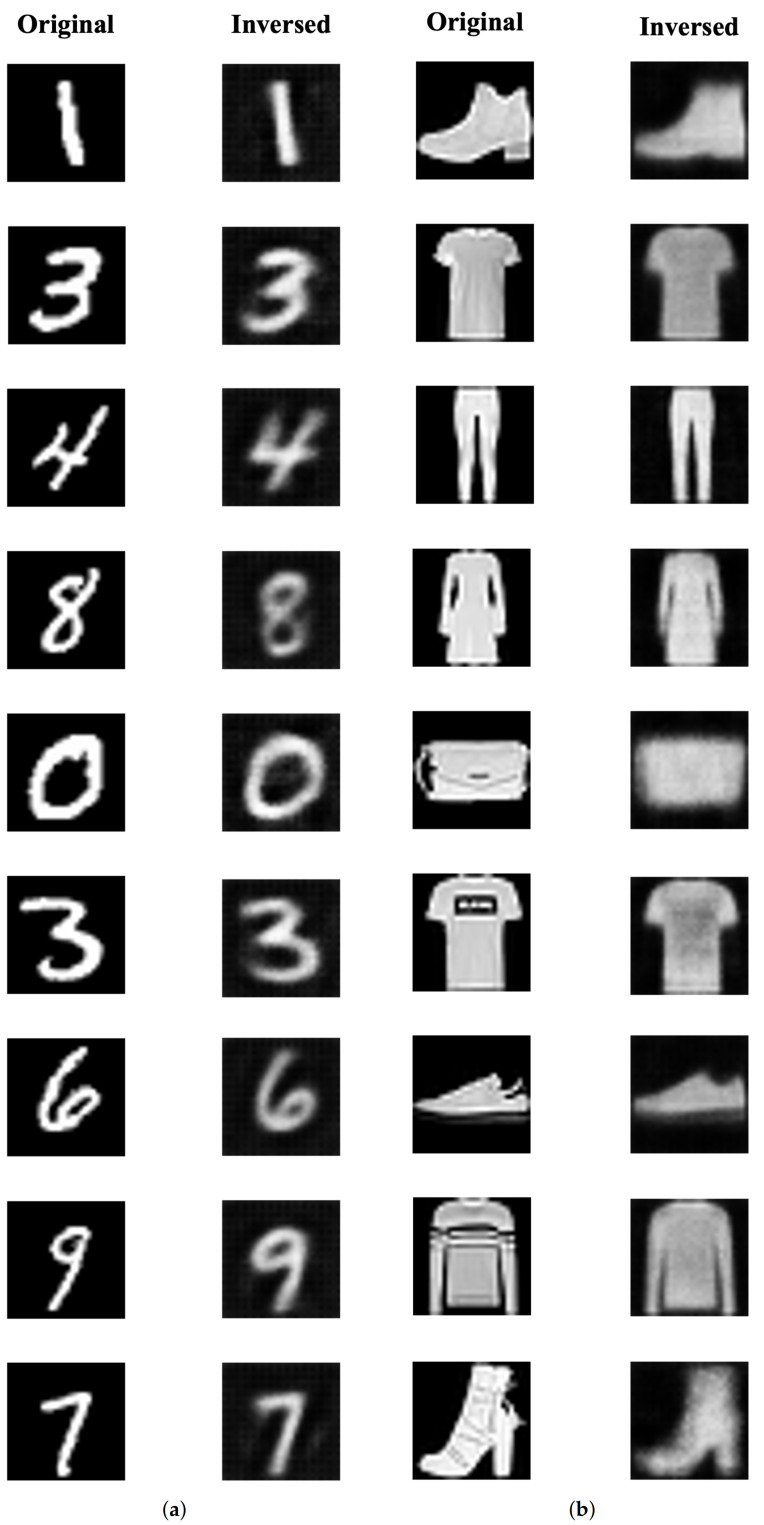
Reconstruction results on MNIST and Fashion-MNIST datasets. (**a**) MNIST dataset. (**b**) Fashion-MNIST dataset.

**Figure 5 sensors-21-07806-f005:**
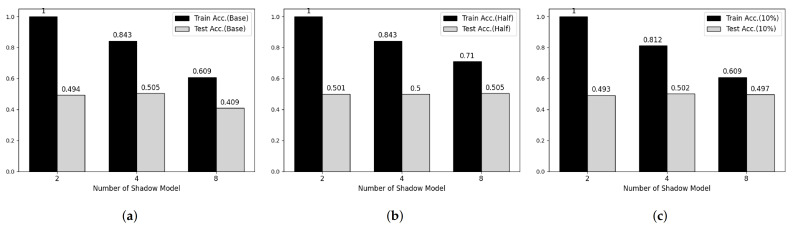
Membership inference attack results on MNIST datasets. (**a**) Base Model. (**b**) Half-data trained Model. (**c**) 10%-data trained Model.

**Figure 6 sensors-21-07806-f006:**
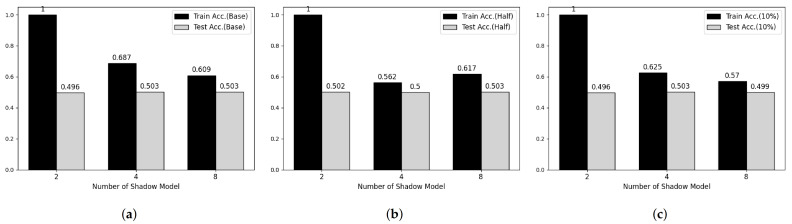
Membership inference attack results on Fashion-MNIST datasets. (**a**) Base Model. (**b**) Half-data trained Model. (**c**) 10%-data trained Model.

### 5.4. Membership Inference Attack

To verify the feasibility of our third attack model, we performed experiments on two main parameters, which are the number of shadow models and the degree of overfitting of target model. Figure 5 and Figure 6 shows partial results of our experiments showing somewhat tendency on MNIST and Fashion-MNIST dataset, respectively. First, considering the effect of the number of shadow models, the attack model shows overfitted manner when the number of shadow model is 2 in both datasets. However, according to the number of shadow models is increasing, the train accuracy of attack model is decreasing and there was no meaningful difference after 8. Secondary, according to Shokri et al., overfitting is not the only factor in vulnerability to membership inference attack, but also it is an important factor [6]. Therefore, we trained both target models using half and 20% of the training data, respectively, and performed membership inference attack. However, we cannot observe any difference with our first consideration.

Table 6 and Table 7 shows whole results of our experiments. As similar to Figure 5 and Figure 6, the recall and precision of training phase, also, decreased according to the number of shadow models is increasing and the recall and precision of test phase are close to 0.5 meaning that the attack model cannot recognize membership information of test datasets.

It seems that such results caused from range of output space. Since BGV/BFV and CKKS schemes, which are supported by He-transformer, do not support division operation, the model encrypted with such schemes cannot use softmax or other logit functions as output activation function. Thus, the range of output space is infinite and each shadow model has totally different output space. As a result, HE-based deep learning model shows resistance to membership inference attack.

## 6. Conclusions

As HE enables arithmetic operations on encrypted data and provides quantum resistance, it is considered as a countermeasure of attacks on deep learning models. However, when HE-based deep learning model is deployed as a service, there are uncovered areas by HE schemes, such as communication link between client and trusted party, input and output data and attacks from insider. In this paper, we propose three attack models that targeting each of the uncovered areas using adversarial example, reconstruction attack and membership inference attack, respectively. We also show two attack models exploiting communication link between client and trusted party using adversarial example and input and output data using reconstruction attack is feasible in our experiments.

Each attack model already has countermeasures, such as adversarial learning or denoising for the adversarial example, and differentially private machine learning for the privacy attacks. However, since additional computation required from such countermeasures is too expensive in current HE schemes, such countermeasures are infeasible. Thus, to adopt HE to real service application, it seems that the practical multi-key based HE schemes or a new methods to decrease the computational cost used computing countermeasure is required.

## Figures and Tables

**Figure 1 sensors-21-07806-f001:**
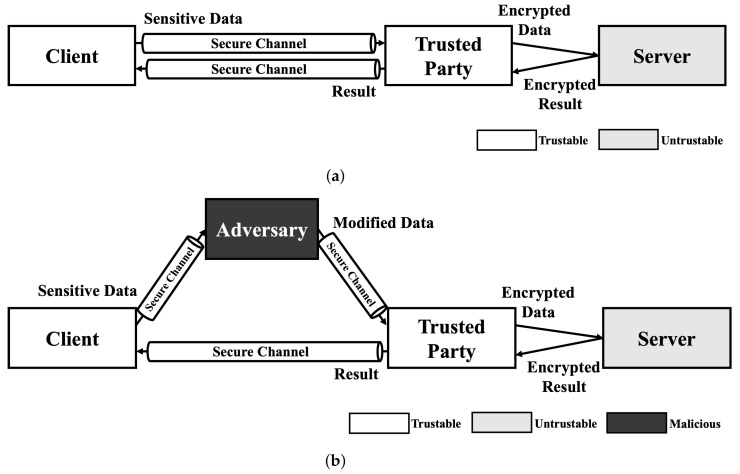
Service scenario of HE deep learning model and exploit scenario using adversarial example. (**a**) Normal scenario of HE-based deep learning model service. (**b**) Adversarial attack scenario using Man-In-The-Middle (MITM) attack.

**Figure 2 sensors-21-07806-f002:**
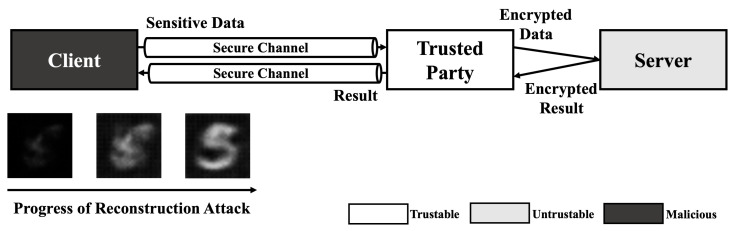
Reconstruction attack scenario on HE-based deep learning model.

**Figure 3 sensors-21-07806-f003:**
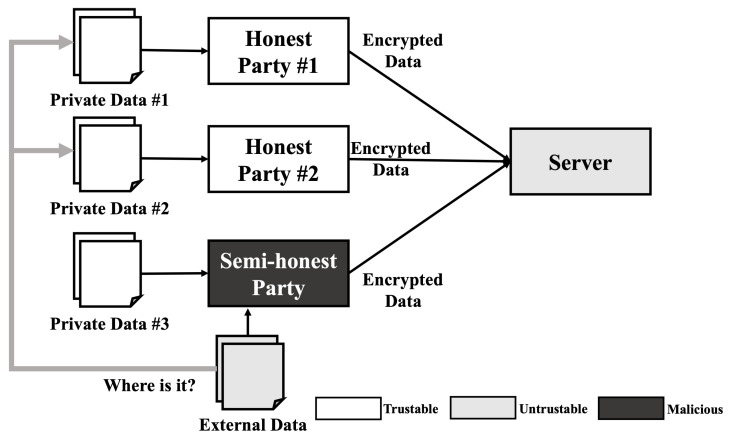
Membership inference attack scenario on HE-based deep learning model.

**Table 1 sensors-21-07806-t001:** Target model architecture for MNIST dataset.

Layer	Description	# of Parameters
Convolution Layer	2D Convolution layer which has 5 output filters with (5, 5) kernel and (2, 2) stride	130
Square Activation	Squares each of the outputs of previous layer	0
Flatten	Reshape 2D outputs of previous layer to 1D	0
Dense-100	Dense layer with 100 node	98,100
Square Activation	Squares each of the outputs of previous layer	0
Dense-10	Output layer consists of dense layer with 10 node	1010

**Table 2 sensors-21-07806-t002:** Target model architecture for Fashion-MNIST dataset.

Layer	Description	# of Parameters
Convolution Layer	2D Convolution layer which has 5 output filters with (5, 5) kernel and (2, 2) stride	130
Flatten	Reshape 2D outputs of previous layer to 1D	0
Dense-128	Dense layer with 128 node	125,568
Square Activation	Squares each of the outputs of previous layer	0
Dense-10	Output layer consists of dense layer with 10 node	1280

**Table 3 sensors-21-07806-t003:** Baseline accuracy of target models.

Model	Train	Test
**MNIST**	0.990	0.971
**Fashion-MNIST**	0.934	0.884

**Table 4 sensors-21-07806-t004:** Adversarial attack results on baseline target models.

	Adversarial Attack Method			
Model	None	FGSM	BIM	C&W
**MNIST**	0.971	0.081	0.017	0.027
**Fashion-MNIST**	0.884	0.094	0.013	0.031

**Table 5 sensors-21-07806-t005:** Reclassification accuracy and SSIM of reconstruction attack results.

Model	Base-Line Acc.	Re-Classification Acc.	SSIM
**MNIST**	0.999	0.945	0.998
**Fashion-MNIST**	0.941	0.832	0.972

**Table 6 sensors-21-07806-t006:** Membership inference attacks results on MNIST dataset.

	Model	Base Model			Half Data			20% Data		
# Shadow	Phase	Acc.	Rec.	Pre.	Acc.	Rec.	Pre.	Acc.	Rec.	Pre.
**2**	**Train** **Test**	1.00 0.494	1.00 0.498	1.00 0.494	1.00 0.501	1.00 0.395	1.00 0.502	1.00 0.493	1.00 0.695	1.00 0.495
**4**	**Train** **Test**	0.843 0.505	0.750 0.257	0.923 0.510	0.843 0.500	0.937 0.731	0.789 0.500	0.812 0.502	0.937 0.823	0.750 0.501
**8**	**Train** **Test**	0.609 0.498	0.656 0.620	0.600 0.498	0.710 0.505	0.578 0.388	0.787 0.506	0.609 0.497	0.703 0.704	0.592 0.498

**Table 7 sensors-21-07806-t007:** Membership inference attacks results on Fashion-MNIST dataset.

	Model	Base Model			Half Data			20% Data		
# Shadow	Phase	Acc.	Rec.	Pre.	Acc.	Rec.	Pre.	Acc.	Rec.	Pre.
**2**	**Train** **Test**	1.00 0.496	1.00 0.621	1.00 0.497	1.00 0.502	1.00 0.732	1.00 0.501	1.00 0.496	1.00 0.573	1.00 0.496
**4**	**Train** **Test**	0.687 0.503	0.812 0.653	0.650 0.502	0.562 0.500	0.250 0.227	0.666 0.501	0.625 0.503	0.937 0.750	0.576 0.502
**8**	**Train** **Test**	0.609 0.503	0.578 0.570	0.616 0.503	0.617 0.503	0.515 0.524	0.647 0.503	0.570 0.499	0.828 0.860	0.546 0.499

## Data Availability

The data presented in this study are openly available in zalandoresearch/fashion-mnist, reference number [36].

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
