# Peer review of "Is Homomorphic Encryption-Based Deep Learning Secure Enough?"

_sensors, 2021, doi:10.3390/s21237806_

Round 1

Reviewer 1 Report

The paper presents different attack models are proposed on the homomorphic encryption-based deep learning services. The paper is well-structured, while it’s hard to follow due to having long sentences. Furthermore, several concerns should be addressed by authors before accepting the paper:

  • The abstract needs to be revised. Some results using these models should be reported. A brief explanation about the platform/application of these methods would also be helpful in the abstract.
  • Recently, there have been some works introducing different attacks, such as https://doi.org/10.1007/978-981-15-8083-3_3. What is the difference between your model and these works since you claim you propose the first work?
  • The difference between this work and [37] should be clarified.
  • Section 2 can be summarized. However, your proposed model can be described in more detail.
  • The comparison between the presented attack and the recent works (as mentioned) would be helpful.
  • Considering the quantum computer treat to break the classical-based architecture, how can the presented algorithm be integrated with the current PQC scheme as a hybrid solution?
  • A figure to show the success rate of the attack would be helpful.
  • The references are not up to date.

Author Response

General Revisions to the Paper

Is Homomorphic Encryption-based Deep Learning Secure Enough

by Jinmyeong Shin, Seok-Hwan Choi and Yoon-Ho Choi

Responses to the individual comments of each reviewer appear in subsequent pages. To distinguish reviewers’ comments from our replies, we write the reviewers’ comments in bold roman and our replies in italics (response) and blue roman (changes or supplemental material). Please note that in the reviewers’ comments section and page numbers refer to the old version of the paper whereas in our replies they refer to the new version of the paper.

Reviewer #1:

Thanks for your comments again. In this revision, we sincerely addressed your all the concerns. Please read our response carefully.

  1. Reviewer’s comment: The abstract needs to be revised. Some results using these models should be reported. A brief explanation about the platform/application of these methods would also be helpful in the abstract.

Author’s response: We appreciate your endeavor and helpful comments to improve our paper. Following your comment, we added a brief explanation about the real world application and numerical results of our experiment in abstract.

Changes made in abstract:  

(Line 15 to 16) In addition, each attack description describes real-world exploit scenarios for financial and medical services.

(Line 16 to 19) we show that the adversarial example, which decreased average classification accuracy from 0.927 to 0.043, and reconstruction attacks, which showed average re-classification accuracy of 0.888, are a practical threat to homomorphic encryption-based deep learning models.

  1. Reviewer’s comment: Recently, there have been some works introducing different attacks, such as https://doi.org/10.1007/978-981-15-8083-3_3. What is the difference between your model and these works since you claim you propose the first work?

Author’s response: We appreciate for helpful comments to improve our paper. Following your comment, we supplemented https://doi.org/10.1007/978-981-15-8083-3_3 in section 2.4 and addressed the difference between https://doi.org/10.1007/978-981-15-8083-3_3 in section 2.4 and our method.

Changes made in section 2.4: Although the HE schemes and HE-based deep learning model have been recently improved, it is difficult to define practical service scenarios due to lack of some functionality and performance. Thus, defining practical attack models for specific service scenario using He-based deep learning model is difficult. However, when considering performance of He-based deep learning, the attack model can be defined easily.

In this perspective, Wu et al. [37] proposed an attack method for He-based deep learning. In this method, the adversary must be the participant of secure multi-party computation(sMPC) using HE. When an adversary trains a model for the first time, the adversary saves the current model parameters. After processing a few steps, the adversary exploits the model by uploading the difference of current model parameters and saved model parameters. Such attack can easily reduce the accuracy of the model, but such reduced accuracy can be easily found before deployed as service. Therefore, the method of Wu [37] is not practical threat to services using HE-based deep learning models.

  1. Reviewer’s comment: The difference between this work and [37] should be clarified.

Author’s response: Thank you for your comment. Following your comment, we supplemented difference between [37] and our method in detail in section 4.1.3

Changes made at line 301 to 302, section 4.1.3: So, we slightly changed the attack method by removing inter-layer feature extraction step as shown in Algorithm 1.

  1. Reviewer’s comment: Section 2 can be summarized. However, your proposed model can be described in more detail.

Author’s response: We appreciate for helpful comments to improve our paper. Following your comment, we supplemented summarization of section 2 and specific examples of our attack model in section 3

Changes made at line 193 to 200, section 2: Different from the existing attacks that can be detected before service deployment, we propose three attacks that consider the vulnerabilities of the service scenario; (1) an adversarial attack exploiting communication link between client and trusted party; (2) a reconstruction attack using the paired input and output data; and (3) a membership inference attack by malicious insider. The first two attacks exploit services using HE-based deep learning which is already deployed. The last attack, membership inference attack, doesn't interact with other parties during attack phase, making it difficult to detect the attack before the target system is exploited.

Changes made at line 247 to 251 and line 271 to 278, section 3:

For example, when an adversary performs Fredrikson et al .'s black-box reconstruction attack [5], the adversary sends an arbitrary image to the server and gets result from the server. After comparing difference between target label and result, the image is updated using gradient descent method. By repeating such steps, the adversary can reconstruct an image of the target label.

For example, when a semi-honest adversary performs Shokri et al.'s membership inference attack [6], the adversary can construct the ideal shadow model because model parameters and architecture are already shared to all parties. After constructing the shadow model, the adversary trains the shadow model and an attack model using their own data. Then, the adversary can analyze the classification result of arbitrary data to perform a membership inference attack. Also, the membership inference attack performed by the adversary cannot be detected since there is no interaction with other parties.

  1. Reviewer’s comment: The comparison between the presented attack and the recent works (as mentioned) would be helpful.

Author’s response: Thank you for your comment. However, there is a mutually exclusive condition to compare our work and the recent work. To compare these two works, we must define same attack condition such as target scenario and target system. But, because of the difference between our work targeting deployed services and the recent work targeting the deep learning model itself during training phase, we cannot construct same attack condition for comparison. Instead, we supplemented an explanation of the difference between our work and the recent work in section 2.4

Changes made at section 2.4: Although the HE schemes and HE-based deep learning model have been recently improved, it is difficult to define practical service scenarios due to lack of some functionality and performance. Thus, defining practical attack models for specific service scenario using He-based deep learning model is difficult. However, when considering performance of He-based deep learning, the attack model can be defined easily.

In this perspective, Wu et al. [37] proposed an attack method for He-based deep learning. In this method, the adversary must be the participant of secure multi-party computation(sMPC) using HE. When an adversary trains a model for the first time, the adversary saves the current model parameters. After processing a few steps, the adversary exploits the model by uploading the difference of current model parameters and saved model parameters. Such attack can easily reduce the accuracy of the model, but such reduced accuracy can be easily found before deployed as service. Therefore, the method of Wu [37] is not practical threat to services using HE-based deep learning models.

Different from the existing attacks that can be detected before service deployment, we propose three attacks that consider the vulnerabilities of the service scenario; (1) an adversarial attack exploiting communication link between client and trusted party; (2) a reconstruction attack using the paired input and output data; and (3) a membership inference attack by malicious insider. The first two attacks exploit services using HE-based deep learning which is already deployed. The last attack, membership inference attack, doesn't interact with other parties during attack phase, making it difficult to detect the attack before the target system is exploited.

  1. Reviewer’s comment: Considering the quantum computer treat to break the classical-based architecture, how can the presented algorithm be integrated with the current PQC scheme as a hybrid solution?

Author’s response: Thank you for your comment. However, our work argues that the current HE schemes are not sufficient to adopted in various services even if recent HE schemes have quantum resistance. Therefore, we cannot suggest a hybrid solution by integrating our work with the current PQC scheme, since we already have quantum resistance. Instead, we supplemented a mention about quantum resistance in section 5.

Changes made at line 387 to 388, section 5: Since HE enables arithmetic operations on encrypted data and provides quantum resistance, it is considered as a countermeasure of attacks on deep learning models.

  1. Reviewer’s comment: A figure to show the success rate of the attack would be helpful.

Author’s response: We appreciate your endeavor and helpful comments to improve our paper. Since the utility of reconstructed data is dependent to adversaries’ final goal, the reconstruction attack has ambiguity to define explicit success rate, instead we supplemented extra experiments to show the utility of reconstructed data is very high and it is threatening in section 4.3 and Table 5

Changes made at line 345 to 354, section 4.3: To show the overall attack success rate, we measured re-classification accuracy, the classification accuracy of reconstructed data which used in Fredrikson et al.'s reconstruction attack [5], and Structural Similarity(SSIM) which calculates the similarity between the original image and the reconstructed image [42]. Table 5 shows measured reconstruction results of each model. The base-line accuracy refers to the classification accuracy of the original images that is the target of reconstruction attack. As shown in Table 5, the re-classification accuracy shows decreased from 0.999 and 0.941 to 0.945 and 0.832 for each dataset, respectively. However, the average SSIM of the reconstructed image showed impressive scores of 0.998 and 0.972, respectively.

Changes made at Table 5:

Reviewer 2 Report

The authors raised a question about the security of  the using homomorphic encryption in deep learning. They then considered three attack models which are the adversarial attack, the reconstruction attack and the membership inference attack to specify and verify the feasibility of exploiting possible security vulnerabilities.

The topic and approach presented and novel and can be considered as perspective and interesting for the readers. I would suggest accepting this paper with some minor modifications:

  1. Please add some related works in which the security and privacy of homomorphic encryption are affected by malicious attacks. 
  2. Do you have numerical results such as some tables for the reconstruction attack?
  3. Please check the English writing in the following lines 21,22,67,73,97,230,265,342.

Author Response

General Revisions to the Paper

Is Homomorphic Encryption-based Deep Learning Secure Enough?

by Jinmyeong Shin, Seok-Hwan Choi and Yoon-Ho Choi

Responses to the individual comments of each reviewer appear in subsequent pages. To distinguish reviewers’ comments from our replies, we write the reviewers’ comments in bold roman and our replies in italics (response) and blue roman (changes or supplemental material). Please note that in the reviewers’ comments section and page numbers refer to the old version of the paper whereas in our replies they refer to the new version of the paper.

Reviewer #2:

Thanks for your comments again. In this revision, we sincerely addressed your all the concerns. Please read our response carefully.

  1. Reviewer’s comment: Please add some related works in which the security and privacy of homomorphic encryption are affected by malicious attacks. 

Author’s response: We appreciate your endeavor and helpful comments to improve our paper. Following your comment, we supplemented an existing related work where the security of homomorphic encryption-based deep learning is affected by malicious attacks in section 2.4

Changes made in section 2.4:  Although the HE schemes and HE-based deep learning model have been recently improved, it is difficult to define practical service scenarios due to lack of some functionality and performance. Thus, defining practical attack models for specific service scenario using He-based deep learning model is difficult. However, when considering performance of He-based deep learning, the attack model can be defined easily.

In this perspective, Wu et al. [37] proposed an attack method for He-based deep learning. In this method, the adversary must be the participant of secure multi-party computation(sMPC) using HE. When an adversary trains a model for the first time, the adversary saves the current model parameters. After processing a few steps, the adversary exploits the model by uploading the difference of current model parameters and saved model parameters. Such attack can easily reduce the accuracy of the model, but such reduced accuracy can be easily found before deployed as service. Therefore, the method of Wu [37] is not practical threat to services using HE-based deep learning models.

Different from the existing attacks that can be detected before service deployment, we propose three attacks that consider the vulnerabilities of the service scenario; (1) an adversarial attack exploiting communication link between client and trusted party; (2) a reconstruction attack using the paired input and output data; and (3) a membership inference attack by malicious insider. The first two attacks exploit services using HE-based deep learning which is already deployed. The last attack, membership inference attack, doesn't interact with other parties during attack phase, making it difficult to detect the attack before the target system is exploited.

  1. Reviewer’s comment: Do you have numerical results such as some tables for the reconstruction attack?

Author’s response: We appreciate your endeavor and helpful comments to improve our paper. Following your comment, we supplemented extra experiments to show the utility of reconstructed data is very high and it is threatening in section 4.3 and Table 5

Changes made at line 345 to 354, section 4.3: To show the overall attack success rate, we measured re-classification accuracy, the classification accuracy of reconstructed data which used in Fredrikson et al.'s reconstruction attack [5], and Structural Similarity(SSIM) which calculates the similarity between the original image and the reconstructed image [42]. Table 5 shows measured reconstruction results of each model. The base-line accuracy refers to the classification accuracy of the original images that is the target of reconstruction attack. As shown in Table 5, the re-classification accuracy shows decreased from 0.999 and 0.941 to 0.945 and 0.832 for each dataset, respectively. However, the average SSIM of the reconstructed image showed impressive scores of 0.998 and 0.972, respectively.

Changes made at Table 5:

  1. Reviewer’s comment: Please check the English writing in the following lines 21,22,67,73,97,230,265,342.

Author’s response: Thank you for your comment. After we carefully overview the paper several times, we corrected grammatical errors that you mentioned.

Round 2

Reviewer 1 Report

The response letter does not address my concerns thoroughly. The difference between this work and the state-of-the-art should be discussed, and the advantages and disadvantages should be explained.

Author Response

General Revisions to the Paper

Is Homomorphic Encryption-based Deep Learning Secure Enough?

by Jinmyeong Shin, Seok-Hwan Choi and Yoon-Ho Choi

To reflect entire reviewer’s second round opinions, we revised our paper thoroughly and we concluded to respond to all comments including first round.

Responses to the individual comments of each reviewer appear in subsequent pages. To distinguish reviewers’ comments from our replies, we write the reviewers’ comments in bold roman and our replies in italics (response) and blue roman (changes or supplemental material). Please note that in the reviewers’ comments section and page numbers refer to the old version of the paper whereas in our replies they refer to the new version of the paper.

In addition, the changes made in our first revision highlighted in green and the changes made in our second revision highlighted in blue on our manuscript file.

Thanks for your comments again. In this revision, we sincerely addressed your all the concerns. Please read our response carefully.

1st Round Comments:

Reviewer #1:

  1. Reviewer’s comment: The abstract needs to be revised. Some results using these models should be reported. A brief explanation about the platform/application of these methods would also be helpful in the abstract.

Author’s response: We appreciate your endeavor and helpful comments to improve our paper. Following your comment, we revised our abstract to address specific field of real-world application and supplemented brief numerical results of our experiments.

Changes made in abstract:  As the amount of data collected and analyzed by machine learning technology increases, data that can identify individuals is also being collected in large quantities. In particular, as deep learning technology, which requires a large amount of analysis data, is activated in various service fields, the possibility of exposing sensitive information of users increases, the user privacy problem is growing more than ever. As a solution to this user's data privacy problem, homomorphic encryption technology, which is an encryption technology that supports arithmetic operations using encrypted data, has been applied to various field including finance and health care in recent years. If so, is it possible to use the deep learning service while preserving the data privacy of users by using the data to which homomorphic encryption is applied? In this paper, we propose three attack methods to infringe user's data privacy by exploiting possible security vulnerabilities in the process of using homomorphic encryption-based deep learning services for the first time. To specify and verify the feasibility of exploiting possible security vulnerabilities, we propose three attacks: (1) an adversarial attack exploiting communication link between client and trusted party; (2) a reconstruction attack using the paired input and output data; and (3) a membership inference attack by malicious insider. In addition, we describe real-world exploit scenarios for financial and medical services. From the experimental evaluation results, we show that the adversarial example and reconstruction attacks are a practical threat to homomorphic encryption-based deep learning models. The adversarial attack decreased average classification accuracy from 0.927 to 0.043, and the reconstruction attack showed average re-classification accuracy of 0.888 respectively.

  1. Reviewer’s comment: Recently, there have been some works introducing different attacks, such as https://doi.org/10.1007/978-981-15-8083-3_3. What is the difference between your model and these works since you claim you propose the first work?

Author’s response: We appreciate for helpful comments to improve our paper. Following your comment, we supplemented https://doi.org/10.1007/978-981-15-8083-3_3 in section 3.2 and addressed the difference between https://doi.org/10.1007/978-981-15-8083-3_3 and our method. The main difference is as follows. The attack you mentioned targets the training phase of HE-based deep learning model. Therefore, the attacks can interrupt the service provider before service is deployed, but cannot exploit the HE-based deep learning service after the service is deployed. Different from such attack, our attack model targets already deployed services that can spill sensitive information of users directly when the service is exploited. To explain this difference, we supplemented descriptions on the existing attacks and our approach in section 3.2

Changes made in line 185 to 201, section 3.4:

Wu et al. [37] proposed an attack method for He-based deep learning. In this method, the adversary must be the participant of federated learning using HE. When an adversary trains a model for the first time, the adversary saves the current model parameters. After processing a few steps, the adversary exploits the model by uploading the difference between current model parameters and saved model parameters to reset the training results of other parties. Such attack can easily reduce the accuracy of the model, but the reduced accuracy can be easily found before deployed as service. Therefore, their method is not practical threat to services using HE-based deep learning models.

Different from the existing attacks that cannot be applied to HE-based deep learning model or that can be detected before service deployment, we propose three attacks that consider the vulnerabilities of the practical HE-based deep learning service scenario. The first two attacks exploit services using HE-based deep learning which is already deployed. In such scenarios, an adversary can exploit target model without access to the key to decrypt data  which is encrypted using the HE scheme. The last attack, membership inference attack, doesn't interact with other parties during attack phase, making it difficult to detect the attack before the target system is exploited.

  1. Reviewer’s comment: The difference between this work and [37] should be clarified.

Author’s response: Thank you for your comment. To address your comment, we rewrote section 5.1.3. Specifically, we described the difference of assumption to exploit model and clarified the procedural difference in detail.

Changes made at section 5.1.3: Reconstruction attack is implemented based on He et al. 's black-box reconstruction attack [21]. To extract inter-layer features, He et al.'s attack assume collaborative inference model that each layer of model is computed in different environment. However, since all inter-layer calculation results are encrypted, the adversary cannot extract inter-layer features from the model. Considering such limitation, we modified He et al. 's attack method to using the output of target model only and the modified version of reconstruction attack is shown in Algorithm 1. The function BlackBoxAttack() shows overall operation of black-box reconstruction attack. First, the training dataset for generator is set to a dataset which has similar distribution with training dataset of the target model(line 4). Then, generator G is trained with these dataset(line 5). After training of generator is done, generator G generates output x^0 which is almost equivalent to original data x0. Most of generator training process(line 9-18) is similar to the He et al. 's method. However, to replace inter-layer features with the output of target model, we modified the loss calculation method as shown in line 13.

  1. Reviewer’s comment: Section 2 can be summarized. However, your proposed model can be described in more detail.

Author’s response: We appreciate for helpful comments to improve our paper. Following your comment, we summarized section 2 by removing some outdated references and modifying descriptions related to homomorphic encryption. Also, to describe our method in more detail, we supplemented specific attack exmaples in proposed attack models.

Changes made at line 130 to 147, section 2: The latest HE scheme called Fully Homomorphic Encryption(FHE) was first introduced by C. Gentry [24]. Gentry's FHE scheme removes the limitation of the number of operations by introducing bootstrapping operation. Based on Gentry's FHE scheme, recent FHE schemes, such as Brakerski-Gentry-Vaikuntanathan(BGV [25], Brakerski-Fan Vercauteren-Vaikuntanathan(BFV) [26] and Cheon-Kim-Kim-Song(CKKS) [27], have improved their performance to a practical level by optimizing the lattice structure and operations for the Ring Learning with Error(RLWE) problem. In addition, since recent FHE schemes, including Gentry's FHE scheme, are designed based on the lattice encryption scheme considering the quantum computing environment, modern FHE schemes are considered as strong candidates for sustainable privacy-preserving computation technique after the quantum computing era.

Changes made at line 248 to 252 and line 272 to 279, section 4:

For example, when an adversary performs Fredrikson et al .'s black-box reconstruction attack [5], the adversary sends an arbitrary image to the server and gets result from the server. After comparing difference between target label and result, the image is updated using gradient descent method. By repeating such steps, the adversary can reconstruct an image of the target label.

For example, when a semi-honest adversary performs Shokri et al.'s membership inference attack [6], the adversary can construct the ideal shadow model because model parameters and architecture are already shared to all parties. After constructing the shadow model, the adversary trains the shadow model and an attack model using their own data. Then, the adversary can analyze the classification result of arbitrary data to perform a membership inference attack. Also, the membership inference attack performed by the adversary cannot be detected since there is no interaction with other parties.

  1. Reviewer’s comment: The comparison between the presented attack and the recent works (as mentioned) would be helpful.

Author’s response: Thank you for your comment. Since the exploit point of the attack, you mentioned, and our attack model is different, it is hard to compare numerical results of each attack. Instead, we supplemented descriptions to show the difference between the recent attack and our attack models. Also, we supplemented comparison with some more recent works exploiting privacy-preserving deep learning models in section 3.2.

Changes made at section 3.2: As many privacy-preserving deep learning methods are introduced, some researchers have proposed attack methods targeting privacy-preserving deep learning models to verify the security of the existing privacy-preserving deep learning or to show inadequacy its as a service.

Rahman et al. [33] tried to exploit differential privacy based privacy-preserving deep learning model using membership inference attack proposed by Shokri et al. [6]. They showed that differential privacy based stochastic gradient descent method decreases the attack success rate meaningfully. However, since their attack scenario requires model information such as model structure and learning parameters, this scenario is not applicable to HE-based deep learning models where the adversary cannot acquire model information.

Chang et al. [34] proposed reconstruction attack methods targeting machine perceptible image encryption based privacy-preserving deep learning model. Although machine perceptible image encryption results are human imperceptible, the encrypted image contains context of original image to be classified by machines. Thus, Chang et al.proposed reconstruction algorithm that re-arranges the pixels of encrypted image using the context of the original image. However, since recent HE schemes provide semantic security, it is almost impossible to get a general context of the original data from the encrypted data. Therefore, their methods cannot be applied to HE-based deep learning models.

Wu et al. [37] proposed an attack method for He-based deep learning. In this method, the adversary must be the participant of federated learning using HE. When an adversary trains a model for the first time, the adversary saves the current model parameters. After processing a few steps, the adversary exploits the model by uploading the difference between current model parameters and saved model parameters to reset the training results of other parties. Such attack can easily reduce the accuracy of the model, but the reduced accuracy can be easily found before deployed as service. Therefore, their method is not practical threat to services using HE-based deep learning models.

Different from the existing attacks that cannot be applied to HE-based deep learning model or that can be detected before service deployment, we propose three attacks that consider the vulnerabilities of the practical HE-based deep learning service scenario. The first two attacks exploit services using HE-based deep learning which is already deployed. In such scenarios, an adversary can exploit target model without access to the key to decrypt data  which is encrypted using the HE scheme. The last attack, membership inference attack, doesn't interact with other parties during attack phase, making it difficult to detect the attack before the target system is exploited.

  1. Reviewer’s comment: Considering the quantum computer treat to break the classical-based architecture, how can the presented algorithm be integrated with the current PQC scheme as a hybrid solution?

Author’s response: Thank you for your comment. However, our work argues that the current HE schemes are not sufficient to adopted in various services even if recent HE schemes have quantum resistance. Therefore, we cannot suggest a hybrid solution by integrating our work with the current PQC scheme, since we already have quantum resistance. Instead, we supplemented a mention about quantum resistance in section 1, 2.2 and 5.

Changes made at line 34 to 38, section 1: Moreover, since modern homomorphic encryption schemes are constructed based on lattice cryptosystem, the homomorphic encryption provides quantum resistance. Therefore, many researchers are endeavoured to apply homomorphic encryption to deep learning as a sustainable countermeasure of attacks for deep learning models

Changes made at line 130 to 147, section 2: The latest HE scheme called Fully Homomorphic Encryption(FHE) was first introduced by C. Gentry [24]. Gentry's FHE scheme removes the limitation of the number of operations by introducing bootstrapping operation. Based on Gentry's FHE scheme, recent FHE schemes, such as Brakerski-Gentry-Vaikuntanathan(BGV [25], Brakerski-Fan Vercauteren-Vaikuntanathan(BFV) [26] and Cheon-Kim-Kim-Song(CKKS) [27], have improved their performance to a practical level by optimizing the lattice structure and operations for the Ring Learning with Error(RLWE) problem. In addition, since recent FHE schemes, including Gentry's FHE scheme, are designed based on the lattice encryption scheme considering the quantum computing environment, modern FHE schemes are considered as strong candidates for sustainable privacy-preserving computation technique after the quantum computing era.

Changes made at line 394 to 395, section 5: Since HE enables arithmetic operations on encrypted data and provides quantum resistance, it is considered as a countermeasure of attacks on deep learning models.

  1. Reviewer’s comment: A figure to show the success rate of the attack would be helpful.

Author’s response: We appreciate your endeavor and helpful comments to improve our paper. To show attack success rate of membership inference attack effectively, we supplemented Figure 5 and Figure 6. Also, descriptions are modified in section 5.4

Changes made at section 5.4: To verify the feasibility of our third attack model, we performed experiments on two main parameters, which are the number of shadow models and the degree of overfitting of target model. Figure 5 and Figure 6 shows partial results of our experiments showing somewhat tendency on MNIST and Fashion-MNIST dataset respectively. First, considering the effect of the number of shadow models, the attack model shows overfitted manner when the number of shadow model is 2 in both datasets. However, according to the number of shadow models is increasing, the train accuracy of attack model is decreasing and there was no meaningful difference after 8. Secondary, according to Shokri et al., overfitting is not the only factor in vulnerability to membership inference attack, but also it is an important factor [6]. Therefore, we trained both target models using half and 20% of the training data respectively and performed membership inference attack. However, we cannot observe any difference with our first consideration.

Table 6 and Table 7 shows whole results of our experiments. As similar to Figure 5 and Figure 6, the recall and precision of training phase, also, decreased according to the number of shadow models is increasing and the recall and precision of test phase are close to 0.5 meaning that the attack model cannot recognize membership information of test datasets.

It seems that such results caused from range of output space. Since BGV/BFV and CKKS schemes, which are supported by He-transformer, do not support division operation, the model encrypted with such schemes cannot use softmax or other logit functions as output activation function. Thus, the range of output space is infinite and each shadow model has totally different output space. As a result, HE-based deep learning model shows resistance to membership inference attack.

Changes made at Figure 5 and Figure 6:

  1. Reviewer’s comment: The references are not up to date.

Author’s response: We appreciate your endeavor and helpful comments to improve our paper. Following your comment, we removed some outdated references such as RSA algorithm and supplemented references about recent attacks that target privacy-preserving deep learning models.

Changes made at section 3.2: Rahman et al. [33] tried to exploit differential privacy based privacy-preserving deep learning model using membership inference attack proposed by Shokri et al. [6]. They showed that differential privacy based stochastic gradient descent method decreases the attack success rate meaningfully. However, since their attack scenario requires model information such as model structure and learning parameters, this scenario is not applicable to HE-based deep learning models where the adversary cannot acquire model information.

Chang et al. [34] proposed reconstruction attack methods targeting machine perceptible image encryption based privacy-preserving deep learning model. Although machine perceptible image encryption results are human imperceptible, the encrypted image contains context of original image to be classified by machines. Thus, Chang et al.proposed reconstruction algorithm that re-arranges the pixels of encrypted image using the context of the original image. However, since recent HE schemes provide semantic security, it is almost impossible to get a general context of the original data from the encrypted data. Therefore, their methods cannot be applied to HE-based deep learning models.

Wu et al. [37] proposed an attack method for He-based deep learning. In this method, the adversary must be the participant of federated learning using HE. When an adversary trains a model for the first time, the adversary saves the current model parameters. After processing a few steps, the adversary exploits the model by uploading the difference between current model parameters and saved model parameters to reset the training results of other parties. Such attack can easily reduce the accuracy of the model, but the reduced accuracy can be easily found before deployed as service. Therefore, their method is not practical threat to services using HE-based deep learning models.

Reviewer #2:

  1. Reviewer’s comment: Please add some related works in which the security and privacy of homomorphic encryption are affected by malicious attacks. 

Author’s response: We appreciate your endeavor and helpful comments to improve our paper. Following your comment, we supplemented an existing related work where the security of homomorphic encryption-based deep learning is affected by malicious attacks in section 2.4

Changes made in section 2.4:  Although the HE schemes and HE-based deep learning model have been recently improved, it is difficult to define practical service scenarios due to lack of some functionality and performance. Thus, defining practical attack models for specific service scenario using He-based deep learning model is difficult. However, when considering performance of He-based deep learning, the attack model can be defined easily.

In this perspective, Wu et al. [37] proposed an attack method for He-based deep learning. In this method, the adversary must be the participant of secure multi-party computation(sMPC) using HE. When an adversary trains a model for the first time, the adversary saves the current model parameters. After processing a few steps, the adversary exploits the model by uploading the difference of current model parameters and saved model parameters. Such attack can easily reduce the accuracy of the model, but such reduced accuracy can be easily found before deployed as service. Therefore, the method of Wu [37] is not practical threat to services using HE-based deep learning models.

Different from the existing attacks that can be detected before service deployment, we propose three attacks that consider the vulnerabilities of the service scenario; (1) an adversarial attack exploiting communication link between client and trusted party; (2) a reconstruction attack using the paired input and output data; and (3) a membership inference attack by malicious insider. The first two attacks exploit services using HE-based deep learning which is already deployed. The last attack, membership inference attack, doesn't interact with other parties during attack phase, making it difficult to detect the attack before the target system is exploited.

  1. Reviewer’s comment: Do you have numerical results such as some tables for the reconstruction attack?

Author’s response: We appreciate your endeavor and helpful comments to improve our paper. Following your comment, we supplemented extra experiments to show the utility of reconstructed data is very high and it is threatening in section 4.3 and Table 5

Changes made at line 345 to 354, section 4.3: To show the overall attack success rate, we measured re-classification accuracy, the classification accuracy of reconstructed data which used in Fredrikson et al.'s reconstruction attack [5], and Structural Similarity(SSIM) which calculates the similarity between the original image and the reconstructed image [42]. Table 5 shows measured reconstruction results of each model. The base-line accuracy refers to the classification accuracy of the original images that is the target of reconstruction attack. As shown in Table 5, the re-classification accuracy shows decreased from 0.999 and 0.941 to 0.945 and 0.832 for each dataset, respectively. However, the average SSIM of the reconstructed image showed impressive scores of 0.998 and 0.972, respectively.

Changes made at Table 5:

  1. Reviewer’s comment: Please check the English writing in the following lines 21,22,67,73,97,230,265,342.

Author’s response: Thank you for your comment. After we carefully overview the paper several times, we corrected grammatical errors that you mentioned.

2nd Round Comments:

Reviewer #1:

  1. Reviewer’s comment: The response letter does not address my concerns thoroughly. The difference between this work and the state-of-the-art should be discussed, and the advantages and disadvantages should be explained.

Author’s response: Following your 1st and 2nd round review comments, we revised the whole paper. Please overview our responses and changes for all your comments in detail above.

Round 3

Author Response

Thanks for your comments again. In this revision, we sincerely addressed your all the concerns. Please read our response carefully.

Academic Editor:

  1. Reviewer’s comment: Too many cited references are from arXiv preprint, which is a clear problem, since those papers were not officially published.

Author’s response: We appreciate your endeavor and helpful comments to improve our paper. Following your comment, we replaced 6 references to published version of the same paper or very similar paper which addresses same problems. However, since some papers such as Explaining and harnessing adversarial examples”, “Somewhat practical fully homomorphic encryption”, “Intel ngraph: An intermediate representation, compiler, and executor for deep learning” and “Technical Report on the CleverHans v2.1.0 Adversarial Examples Library” are de facto standard of each field and thousands of papers are already cited such papers, we concluded not to change these papers. Also, we removed two arXiv papers that was referenced for minor explanation.
